# The power of past performance in multidimensional supplier evaluation and supplier selection: Debiasing anchoring bias and its knock-on effects

**Ricky S. Wong** *

Department of Business Analytics and Systems, University of Hertfordshire, Hatfield, United Kingdom

* r.wong3@herts.ac.uk

**Data Availability Statement:** The data files have been uploaded as supporting information.

**Funding:** The author(s) received no specific funding for this work.

## Abstract

This research examines how anchoring bias affects managers' multi-dimensional evaluations of supplier performance, supplier selection, and the effectiveness of two debiasing techniques. We consider the supplier past performance in one performance dimension as the anchor and investigate whether and how this anchor would have a knock-on effects on evaluating a supplier's performance in other dimensions. We conducted two online experimental studies (Study 1, sample size = 104 and Study 2, sample size = 408). Study 1 adopts a 2 x 1 *(high anchor vs. low anchor)* between-subjects factorial experimental design, and Study 2 is a 3 *(debiasing: no, consider-the-opposite, mental-mapping)* x 2 *(high anchor vs. low anchor)* between-subjects factorial design. The results from Studies 1 and 2 suggest that when a supplier has received a low evaluation score in one dimension in the past, participants assign the same supplier lower scores in the other dimensions compared to a supplier that has obtained a high score in the past. We also find that anchoring has a knock-on effect on how likely participants are to choose the same supplier in the future. Our findings highlight the asymmetric effectiveness of 'consider-the-opposite' and 'mental-mapping' debiasing techniques. This research is the first study that examines how anchoring bias managers' evaluations in a multi-dimensional setting and its knock-on effects. It also explores the effectiveness of two low-cost debiasing techniques. A crucial practical implication is that suppliers' exceptionally good or disappointing past performance affects the judgement of supply managers. Hence, managers should use consider-the-opposite or mental-mapping techniques to debias the effect of high and low anchors, respectively.

## Introduction

In almost all industries, organisations rely on suppliers to support their production system. Buying organisations usually evaluate supplier performance against several dimensions [1–3]. However, recent behavioural operations management (BOM) studies have begun to examine how managers' judgements and decisions deviate from the rationality assumption [4–7].

**Competing interests:** The authors have declared that no competing interests exist.

Undoubtedly, a theoretical research gap is evident. Hald and Ellegaard (2011) [8] and Wong (2021) [2] have underscored the significance of comprehending the cognitive processes that underpin supplier evaluation decisions; without such understanding, identifying theories that effectively explain actual behaviour becomes challenging. Similarly, Fahimnia et al. (2019) [9] and Perera, Fahimnia and Tokar (2020) [10] have advocated for continued exploration within BOM research to elucidate how decision-making in practice is shaped by human judgement.

A burgeoning body of literature in behavioural operations has commenced elucidating the pivotal role of human behaviour in Supply Chain and Operations Management (SCOM) contexts [10–17]. Experimentation allows testing of cause-and-effect relationships and facilitates identification of biases in managers' decision-making processes. For instance, SCOM studies have shed light on human cognitive biases (e.g., attribute framing effect) as the cause of suboptimal order in the newsvendor problem [18,19], and biases in supplier evaluation [2]. A new inventory ordering policy received more favourable evaluation [6], ordering decisions in dual sourcing [20]. However, little is known about anchoring effects of a supplier's past performance in present supplier evaluation. Therefore, this experimental research aims to examine how the past performance of a supplier—as an anchor—affects multidimensional supplier evaluation and supplier selection. In other words, it considers past supplier performance may bias managers' evaluations of recent supplier performance and supplier selection. However, a significant research gap regards how this bias may be mitigated. Different de-biasing techniques are tested in this research.

When evaluating a supplier's performance, a manager is likely to evaluate the supplier considering its past performance. Information about supplier past performance may help the manager complete the evaluation but does not guarantee an unbiased decision. As discussed in the judgement and decision-making literature, a cognitive bias may arise in the presence of previous assessments, known as anchoring [21–24]. Judgements are affected by the anchor value so that the current estimates are typically close to the anchor.

Another line of inquiry has explored anchoring effects on ordering decisions [7] and suboptimal order in the newsvendor game [25]. The presence of anchoring effects has been investigated in real-life managerial settings such as addressing the weight of various dimensions in multidimensional decision-making [26,27], the newsvendor problem [28], and negotiation in a multi-tier supply chain [29]. However, some studies have found minimal impacts or mixed findings [30,31]. No consensus has been reached on the anchoring effect in the supplier evaluation and supplier selection processes, resulting in a significant research gap. Hence, this study considers the power of anchoring in multidimensional supplier evaluation and supplier selection. We examine the likelihood that a supplier's past evaluation of one dimension may also affect subsequent assessments of the same supplier performance in other dimensions. Along with research on supplier selection [1,2,29], we studied in the paradigm of multidimensional decision making.

Examining the prevalent anchoring effects in supplier evaluations is essential, as it strengthens the generalisability of previous research in the field. We will test whether biased evaluation may lead to choosing an underperforming supplier or terminating a relationship with a competent supplier. Hence the second objective of this research is to investigate two low-cost debiasing techniques that may be adopted in small and medium enterprises. Scholars from other disciplines have recently attempted to develop strategies to remove anchoring bias, but the effectiveness of these techniques is still debatable [32–34]. This study examines two different debiasing techniques, 'consider-the-opposite' and 'mental-mapping'.

Anchoring bias in supplier evaluation raises four practical issues: (1) To what extent are managers sensitive to anchoring effects in multidimensional supplier evaluation? (2) Does past performance in a dimension also anchor managers' judgement on supplier performance in

other dimensions? (3) Does anchoring have a knock-on effect on supplier selection? (4) How may anchoring effects be mitigated? The present research addresses these empirical questions by conducting two experimental studies (Studies 1 and 2). Studies 1 and 2 are online controlled experiments investigating the anchoring effects and effectiveness of two alternative debiasing techniques. The study's results have significant theoretical implications and provide valuable insights for practitioners. Specifically, this unique contribution of this research is to offer low-cost decision aids in reducing the effects of low and high anchors on supplier evaluation and supplier selection. Another important contribution is to demonstrate how in multi-dimensional supplier evaluation the past performance in one dimension profoundly affects evaluating suppliers' recent performance.

## Practical relevance

The current research is motivated by direct interactions with the Sales Director of a multi-national toy manufacturer. We interviewed the Director and the Head of the Supply Chain Department in July 2022. Both the Director and the Head discussed their use of multiple sourcing strategy for the modules required for production. Supplier evaluation is one of the most important business processes, because they believed the strong relationship between the quality of suppliers and their product quality. Their observation was that the suppliers' performance whom they had already established a relationship with was not evaluated properly. Specifically, the sourcing and quality assurance team appeared to rely 'too much' on how the suppliers performed in the past. The Director stated "I have seen problems with the suppliers that may not be reflected in supplier evaluations. Also, when the supplier's past performance affected the evaluations, it rendered identifying the suppliers that had made significant improvements difficult. I think that offering training to our colleagues about how to mitigate the biases in their evaluation decisions would be vital to our organisation".

## Theoretical background and hypotheses

### Anchoring and knock-on effects of past performance in multidimensional supplier evaluation and supplier selection

Although this research focuses on how a supplier's past performance in one dimension–an anchor–affects current evaluations and supplier selection, it is paramount to offer the development of anchoring research originated from the psychology and judgement and decision-making fields. Because anchoring effects seem to be ubiquitous in day to day life, The psychology, judgement, and decision-making literature have devoted substantial attention to anchoring effects and their underlying mechanisms [23,30,35,36] and investigating the factors that influence when and how they occur [37–42].

Previous experimental studies have investigated anchoring bias using comparative questions in the quantity estimation of general knowledge [39,42,43]. Participants are typically first asked to assess whether the target answer is higher or lower than an anchor value provided by the experimenter (e.g., "Is the River Nile longer or shorter than 4,000 miles?"). Subsequently, participants are prompted to provide an estimate of the target (i.e., the length of the river) (referred to as the 'direct comparative approach'). Remarkably, anchoring effects persist even in the absence of an explicit comparative question. Merely presenting an anchor and requesting individuals to express their judgment is adequate for inducing anchoring (known as the 'indirect comparative approach') [38,40,42].

To gauge the extent of the anchoring effect, Thornsteinson et al. (2008) [42] compared anchoring effects using the direct comparative approach with those using the indirect

comparative approach. In both experiments, participants evaluated the performance of a hypothetical employee and an instructor in a university setting. The direct comparative approach entailed an initial question such as "Is the employee's/instructor's performance rated a score of 9?" in the high-anchor group (or 1 in the low-anchor group), followed by judgment tasks. Conversely, the indirect comparative approach presented participants with an example rating form displaying the highest (or lowest) score. Notably, both the direct and indirect approaches yielded similar anchoring effects. In the suppliers' performance evaluation context, the indirect comparative approach strengthens the external validity of our research, as supply managers are seldom asked to directly compare whether a supplier's performance is higher (or lower) than a hypothetical value before completing a supplier evaluation.

In addition to manipulating the judgment process (i.e., direct vs. indirect comparison judgments), studies have explored the influence of specific types of anchors, such as relevant versus irrelevant anchors. A relevant anchor provides informative cues that could aid decision-making within the context of interest (e.g., past performance of an employee), whereas an irrelevant anchor offers uninformative cues (e.g., an arbitrary number).

The majority of research indicates that irrelevant anchors impact participants' judgments across various contexts [40,42,44,45]. However, relevant anchors exhibit a similar pattern of influence over judgment and choice behaviours [41,46–50]. Indeed, some researchers have speculated whether the same underlying judgment processes are elicited when relevant or irrelevant anchors are employed [35,45,50].

Anchoring effects have been investigated in the business context, such as the performance appraisal of employees [33], promotion decisions in academia [51], order quantity in a newsvendor problem [10,28], and multi-tier supply chain negotiation [52].

We build our argument and develop the research hypotheses based on various studies focusing on the relationship between anchoring and performance evaluation, although anchoring effects on multidimensional evaluations have *not* been investigated. Some studies have shown that managers' assessments of subordinate performance are affected by an employee's past appraisal scores [33,53]. This phenomenon recalls supplier evaluation. Wong (2021) [2] provides evidence suggesting that managers' evaluation of one dimension in which a supplier performs affects their assessment of the performance of the same suppliers in other dimensions. Overall, this stream of research corroborates the power of the anchor irrespective of whether an anchor is provided externally or self-generated by decision-makers. If the anchoring effect applies to supplier evaluation, a supplier that receives a good score in a previous assessment is likely judged more favourably in the future than a supplier with a poor score. This phenomenon may affect a buying firm's decision-making, thus creating two problems. First, a supplier that has not performed well in the past but then improved its production may not be evaluated fairly and thus not be selected. Second, a supplier with a good track record that falls short in a recent order is likely to be considered favourably. In this case, the decline in supplier performance is more difficult to identify, potentially leading to a product recall.

Our explanation of the underlying mechanism through which past supplier performance anchors managers' judgement is rooted in the scale distortion theory of anchoring [23,24]. This theory posits that the perceived magnitude of a value may be affected by other numerical values presented to managers when judgements are based on the same scale. Experimental studies in which participants make sequential judgements support this theory. For instance, previous evidence shows that the first evaluation (e.g., the assessment of a rabbit's weight) works as an anchor and impacts the subsequent evaluation (e.g., the assessment of a giraffe's weight). More recent research supports this theory by generalising it to other stimuli [21]. Focusing on one dimension of supplier performance, we expect that a high (or low) supplier

score shifts managers' perception of the scale on which supplier performance is assessed. Hence, we propose the following hypothesis:

*Hypothesis 1*: When presented with a high evaluation score of a supplier's past performance in one dimension, participants evaluate the same dimension in current performance more highly than when presented with a low evaluation score in past supplier performance.

This study considers the potential knock-on effect of anchoring on supplier performance evaluations in other dimensions because a supplier's performance is usually evaluated against more than one dimension [1,2]. The scale distortion theory provides insights into how managers' evaluations anchored to a past performance score also affect their evaluations in other dimensions. As predicted, a low or high anchor (i.e., the past performance score in one dimension) distorts the perception of the evaluation scale. When the same evaluation scale is used on other dimensions (e.g., a nine-point evaluation score system), supplier performance evaluations in different dimensions shift toward the lower (or higher) end of the scale, depending on the magnitude of the anchor value.

Our prediction of the knock-on effect is also consistent with a stream of research that examines the halo effect on consumer perception of product safety [54], performance evaluation [53], and other business contexts [55]. This effect suggests that decision-makers allow inferences about performance in a dimension (e.g., creativity at work) to affect evaluations of different dimensions (e.g., interactions with customers and work accuracy). Specifically, a positive evaluation in one dimension leads to favourable evaluations in other dimensions. Applied psychologists contend that performance evaluators are susceptible to bias due to cognitive limitations [35,42]. In addition, decision-makers have a strong tendency to maintain their consistency of judgements, resulting in a homogenous perception of an individual or object [56]. Coupling the distorted perception of the scale with halo effects, we speculate that a high or low anchor value (indexed in the form of past performance in one dimension) impacts managers' evaluations in other dimensions. Hence, we propose the following hypothesis:

*Hypothesis 2*: When presented with a high evaluation score of past supplier performance in one dimension, participants rate other dimensions in current performance more highly than when presented with a low evaluation score in past supplier performance.

The potential anchoring effect on decision-making regarding supplier selection is equally relevant. Supplier selection is a vital component of supply management [57,58], as suppliers play a crucial role in product quality. The relationship between supplier performance evaluation and the choice of supplier allows to predict that the anchoring effects on performance also affect supplier selection choices. Furthermore, evaluation outcomes are crucial to the likelihood that the same buyer-supplier relationship will continue in the future [2]. In the presence of positive evaluations, supply managers have a stronger tendency to engage in future transactions with the supplier. A low-anchor value in past supplier performance (compared to a high anchor) also affects decisions about continuing a relationship with a supplier. Hence, we propose the following hypothesis:

*Hypothesis 3*: When presented with a high evaluation score of past supplier performance in one dimension, participants are more likely to choose the same supplier in the future than when presented with a low evaluation score in past supplier performance.

## Debiasing anchoring effects in supplier evaluation

As debiasing is a relatively new topic in the supply chain management field, no previous studies have investigated the techniques for debiasing anchoring effects in supplier evaluation. Two different approaches exist in the literature on debiasing: modifying the decision-making environment and adapting decision-makers. The former approach aims to reduce anchoring bias by increasing decision-makers' accountability [59] or implementing nudging [33,34]. The

second approach addresses the decision-maker by providing tools to mitigate the effects of bias [33,60]. This study contends that managers rely on heuristics to make decisions with System 1 and make corrections by effortful thinking with System 2 [21,61,62].

This research focuses on the second approach for several reasons. First, previous studies have suggested that this method helps uphold managers' self-esteem [33,63]. Second, changing the decision-making environment is likely to entail investment in new decision support systems and intensive training for managers. This step may not be feasible for small or medium-sized enterprises. Third, the effectiveness of modifying the decision-making environment is often limited. For example, George *et al.* (2000) [32] find that a warning message for prompting decision-makers to use decision support systems does not reduce the anchoring bias. Oz *et al.* (2020) [34] employ digital nudges to debias the anchoring effects on product transfer quantities in a supply-chain context. They show that anchoring effects are reduced only when decision-makers are familiar with the *continued* use of information systems.

The two debiasing techniques examined in this study are 'consider-the-opposite' and 'mental-mapping'. Both techniques are low-cost and based on the psychological mechanism(s) by which anchoring bias affects supplier performance evaluations. The consider-the-opposite technique requires decision-makers to provide reasons against an anchor value. Previous studies have demonstrated the effectiveness of debiasing the anchoring effect with a comparative question in the context of employee performance evaluation [33] and car valuation [64]. This study tests this technique in situations where a comparison question is not used, thus strengthening the method's external validity. Mental remapping is a novel approach, explicitly targeting the distorted evaluation scale due to anchoring. This technique requires decision-makers to map two reference points on an evaluation scale (i.e., the highest and lowest possible scores). This approach has been *only* tested in the sequential anchoring paradigm in a recent study [21]. I have adapted Bahník et al.'s (2019) [21] approach by addressing the anchoring effect without using a comparative question and have tailored it to the multi-dimensional supplier evaluation setting. In this research, the two reference points entail the best (and the worst) supplier, deserving the highest (and lowest) score on the scale. Thus, we test whether these two techniques substantially decrease the anchoring effects on evaluations and supplier selection. Hence, we propose the following hypotheses.

If consider-the-opposite technique helps reduce anchoring effect:

*Hypothesis 4a*: Participants in the low-anchor consider-the-opposite condition give higher evaluation scores than participants in the low-anchor condition.

*Hypothesis 4b*: Participants in the high-anchor consider-the-opposite condition give lower evaluation scores than those in the high-anchor condition.

*Hypothesis 4c*: Participants in the low-anchor consider-the-opposite condition have a higher likelihood of contracting the same supplier in the future than participants in the low-anchor condition.

*Hypothesis 4d*: Participants in the high-anchor consider-the-opposite condition have a lower likelihood of contracting the same supplier in the future than participants in the high-anchor condition.

If mental-mapping technique helps reduce anchoring effect:

*Hypothesis 5a*: Participants in the low-anchor mental-mapping condition assign higher evaluation scores than those in the low-anchor condition.

*Hypothesis 5b*: Participants in the high-anchor mental-mapping condition assign lower evaluation scores than those in the high-anchor condition.

*Hypothesis 5c*: Participants in the low-anchor mental-mapping condition have a higher likelihood of contracting the same supplier in the future than participants in the low-anchor condition.

*Hypothesis 5d*: Participants in the high-anchor mental-mapping condition have a lower likelihood of contracting the same supplier in the future than participants in the high-anchor condition.

## Study 1: Anchoring effects on multidimensional supplier evaluation

### Methods

This study was approved by the University Committee on the Use of Human Subjects with Protocol#: BUS/SF/UH/05903 and by another university in Hong Kong. Participants were presented with an informed consent form prior to the study and were asked to provide written consent.

Study 1 presented a realistic multidimensional supplier evaluation task. Participants were randomly assigned to two conditions: *high-anchor* condition and *low-anchor* condition. We performed a pilot study with master's students to ensure that the sourcing task and information regarding the supplier were easy to understand. To ensure that the descriptions of suppliers' performance were appropriate, we conducted another pilot study with 9 participants who worked for the company at which we interviewed [65] and the interviewees confirmed the validity of the descriptions of suppliers' performance. Participants received a fixed payment of £0.95 for their participation in this study. To incentivise participants' performance, the participants were led to believe that their supplier evaluations would be assessed by two independent judges. They would receive a bonus payment of £0.65 if their evaluations were deemed appropriate. In reality, all the participants were given the bonus payment.

To validate that the outcomes were a consequence of the anchoring manipulation, the sole distinction between the high-anchor and low-anchor conditions resided in the variance of past performance regarding cost. Notably, the past performance of the supplier across the remaining three attributes remained consistent,. Moreover, the information provided and the procedures employed in the current supplier evaluation, encompassing performance across the four attributes, were uniform across both conditions as illustrated in Figs 1 and 2. Therefore, any differences observed could be attributed to the disparate anchor values assigned to the cost attribute in prior supplier evaluations, affirming their anchoring effect.

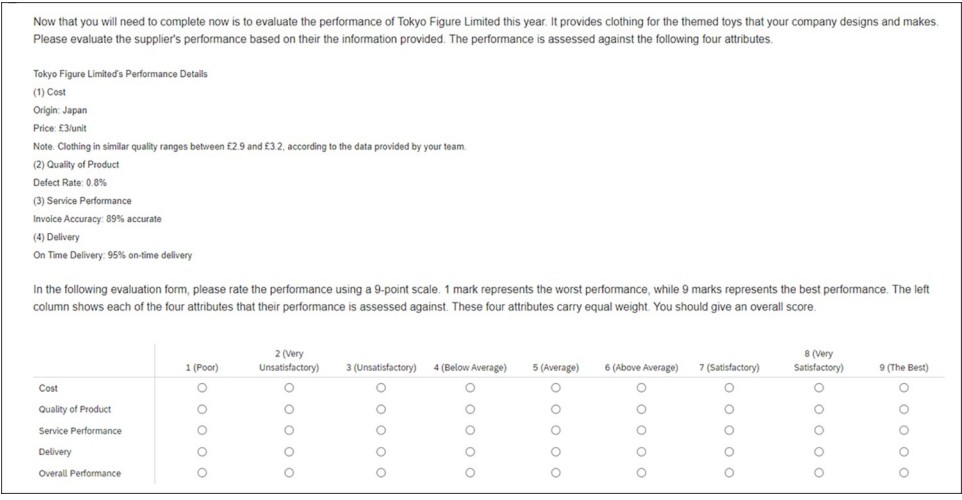

**Fig 1. The multi-dimensional evaluation task of supplier performance.**

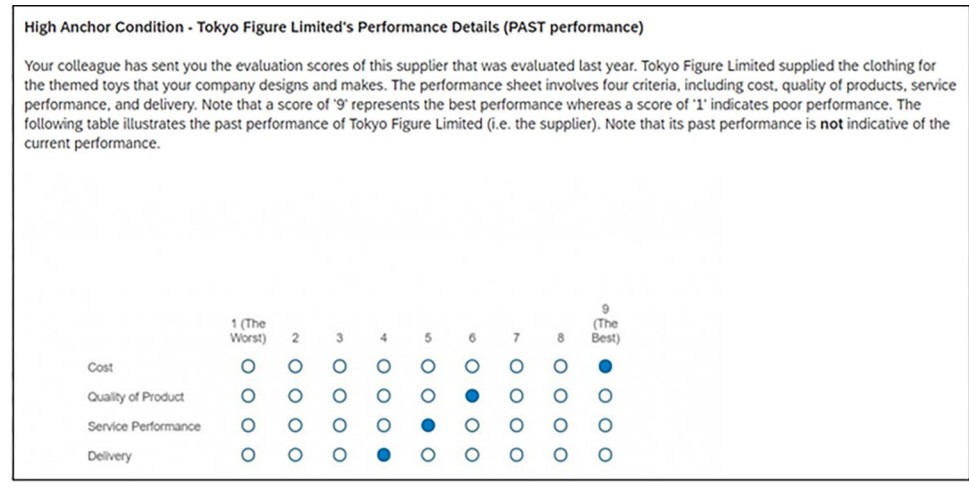

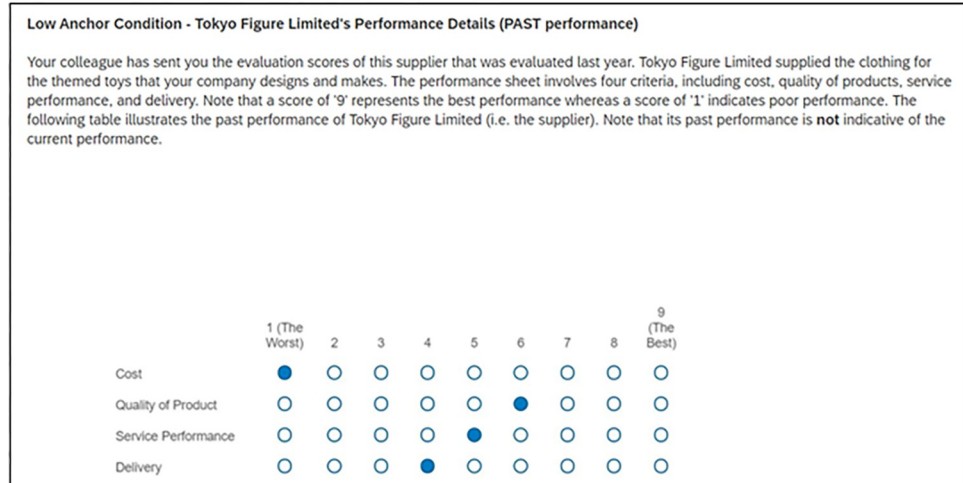

**Fig 2. Anchor manipulation in the high-anchor and low-anchor conditions.**

### Realism checks

I adopted the quantitative approach to evaluate the realism of the scenario and independent variable, ensuring participants would find the scenarios and the corresponding independent variables realistic and understandable [20,65–67]. Three items were used to assess the realism of scenarios and experimental design ('The situation described in the scenario was realistic', 'The scenario was believable', and 'I can imagine myself in the described situation'.) [68]. A seven-point Likert scale was used for all items (1 = strongly disagree, 7 = strongly agree).

A pilot study with 30 employees in the company whom the executives were interviewed was performed for realism checks and to ensure that the evaluation task and suppliers' performance were easy to understand. The three items generated an average score of 5.38 and high internal consistency (Cronbach's alpha = 0.85). The mean scores for the two conditions were very close. The findings show that the participants perceived the scenarios and independent variables as realistic.

### Participants

The *G*Power* software package was utilised to estimate the sample size required, given the experimental design [69]. One hundred and ten participants were recruited in the United

Kingdom from Prolific Academic in 2021. An advantage of this method is that it defines the pool of relevant experienced practitioners (i.e., those with supplier evaluation experience). Multiple participations by the same participant were prevented for maintaining high levels of data quality. Following the ethics guideline of Prolific Academic, any information that could identify individual participants was not collected. Consistent with recommendations for identifying speeders in online studies [70,71], five participants who spent less than one-third of the median completion time were excluded from the study. Additionally, one participant who failed to pass the attention test was removed from the analysis. Consequently, the sample size was reduced to one hundred and four participants (58.70% female). Their mean age was 37.77 years (*S.D.* = 9.63). The participants had an average of 6.88 years of experience evaluating a supplier's performance (s) (*S.D.* = 6.18). There was no time limit for the experiments. The mean completion time was 7.24 minutes (*S.D.* = 9.16).

## Procedure

Firstly, participants read the informed consent form on their computer screen, and they were asked to explicitly indicate whether they agreed to take part in the studies in writing. It was made clear to them that they may withdraw from the study anytime they wished. Participants were then asked to complete demographic questions and indicate their experience in supplier evaluation. They were then provided information about the evaluation task, including the dimensions of the supplier's performance, and they read "*past performance of a supplier did not necessarily indicate the current performance of supplier*". As illustrated in Fig 1, participants were told that the four performance dimensions (i.e., cost, quality, service, and delivery) carried equal weight.

Most importantly, participants were informed that "the four performance dimensions were independent of one another". Prior to the evaluation task, participants were presented with the past performance evaluation conducted one year prior to the current evaluation task (see Fig 2), depending on whether they were assigned to the high-anchor or low-anchor condition. Additionally, they were asked to familiarise themselves with the nine-point scale used in the evaluation task (*1* = poor performance, *9* = best performance).

Before commencing the task, an attention filter question was administered to ensure participants paid attention to the provided information. They were required to indicate their agreement as "strongly agree" in response to the attention filter question. All participants were instructed as follows: "You need to evaluate the most recent performance of Tokyo Figure Limited based on their performance in each of the four attributes: cost, quality of project, service performance, and delivery. You will be provided with information about the supplier's performance in each attribute. Additionally, a nine-point Likert scale is used to complete the evaluation."

First, the anchors were operationalised as the supplier's past performance in the 'cost' dimension. Participants were randomly assigned to two experimental conditions. In the high-anchor condition, participants read a score of '9' in the previous evaluation of the supplier. In the low-anchor condition, the past performance indicator reported a score of '1' in the cost dimension of the past evaluation. In both conditions, participants received identical information about the supplier's past performance in the other three dimensions (i.e., quality of product, service performance and delivery) (see Fig 2 for details). The *only* difference between the two conditions was the past evaluation score on the cost dimension. Then, participants were directed to the evaluation task, and the performance indicators in the four dimensions were presented.

### Dependent measures

**Statistical package and normality check.** All analyses in Studies 1 and 2 were conducted using the Statistical Package for the Social Sciences (SPSS) (Version 29) (see S1 and S2 Files for the full datasets of Studies 1 and 2). The data collection method and the characteristics of independent and dependent measures adhere to most assumptions of independent sample t-tests. Before proceeding with the analyses, it is crucial to assess whether the dependent measures are approximately normally distributed in both the low-anchor and high-anchor conditions. Skewness and kurtosis coefficients were computed for the four evaluation scores. The skewness coefficients ranged from -1.10 to 0.32, while the kurtosis coefficients ranged from -1.02 to 0.80. According to Bryne (2010) [72] and Hair et al. (2010) [73], the normality assumption is met when the skewness coefficient falls between -2.0 and 2.0, and the kurtosis coefficient ranges from -7.0 to +7.0.

*Evaluation scores.* Participants rated the performance of the suppliers on a nine-point scale. A higher score indicates better performance in the relevant dimension.

*Likelihood of using the same supplier (0–100%).* Participants were asked to indicate the likelihood of contracting the same supplier in the future after completing the evaluation task.

### Results

**Anchoring and knock-on effects.** To examine the relationship between the anchor and supplier evaluation, we focused on the evaluation given to the cost dimension and performed an independent sample $t$-test. In line with Hypothesis 1, participants assign a significantly higher score on cost in the high-anchor condition than in the low-anchor condition ($M_{high\ anchor}$ = 7.08 vs $M_{low\ anchor}$ = 3.39; $t$ = 9.78, $df$ = 102, $p < 0.0005$, Cohen's d = 1.94). Thus, Hypothesis 1 is supported. The evaluation scores in other dimensions are higher in the high-anchor condition than in the low-anchor condition. Participants in the high-anchor condition rate supplier performance in terms of service ($M_{high\ anchor}$ = 7.25) higher than those in the low-anchor condition ($M_{low\ anchor}$ = 6.56; $t$ = 2.20, $df$ = 102, $p < 0.05$, Cohen's d = 0.43). Regarding the evaluation of delivery, suppliers in the high-anchor condition receive a significantly higher score ($M_{high\ anchor}$ = 8.38) than those in the low-anchor condition ($M_{low\ anchor}$ = 7.75; $t$ = 2.41, $df$ = 102, $p < 0.05$, Cohen's d = 0.47). In terms of the quality dimension, participants in the high-anchor condition rate the supplier more favourably than those in the low-anchor condition ($M_{high\ anchor}$ = 7.13 vs $M_{low\ anchor}$ = 6.37; $t$ = 2.73, $df$ = 102, $p < 0.01$, Cohen's d = 0.53). Overall, these results support Hypothesis 2.

As predicted by Hypothesis 3, in the high-anchor condition participants indicate a higher likelihood to continue to use the supplier in the future than those in the low-anchor condition ($M_{high\ anchor}$ = 71.29% vs. $M_{low\ anchor}$ = 59.06%; $t$ = 3.27, $df$ = 102, $p < 0.001$, Cohen's d = 0.65). Thus, Hypothesis 3 is supported.

Overall, the results indicates that anchors, in the form of past performance in a dimension, have profound effects on participants' evaluations of supplier performance in other dimensions. Anchors also have a knock-on effect on the likelihood that the same supplier is chosen in the future. Our findings suggest that both evaluation and supplier selection are likely biased when a supplier's past performance in one dimension is at either end of the evaluation scale. Thus, the study's findings highlight the need for efficient techniques to reduce anchoring effects in supplier evaluation and selection.

## Study 2: Relevant anchor of less extreme values and debiasing techniques

This study was approved by the University Committee on the Use of Human Subjects with Protocol#: BUS/SF/UH/05903 and by another university in Hong Kong. Participants were

presented with an informed consent form prior to the study and were asked to provide written consent.

Study 2 was designed to revisit the anchoring and knock-on effects on evaluations and supplier selection by reducing the extremity of the anchors, potentially generalising the findings from Study 1. To this end, Study 2 investigated two debiasing techniques, namely, consider-the-opposite and mental remapping.

## Method

Study 2 was an online 2 (anchor: low or high) × 3 (debiasing: no, consider-the-opposite, mental-mapping) factorial between-subject experiment.

*Participants.* Following the same procedure used to estimate the required sample size using *G*Power* software for Study 2, which was determined to be 398 participants, a total of 420 participants were recruited through the Prolific Academic crowdsourcing platform in 2021. However, eleven participants were identified as speeders, and three participants failed to pass the attention check, utilising the same attention test and methods of identifying speeders as in Study 1. Consequently, 406 participants (54.70% female) were included in the analyses. The mean age of participants was 35.77 years (*S.D.* = 10.04). Each participant received a fixed payment of £1.10. The mean years of participants' experience in supplier evaluation were 6.63 years (*S.D.* = 5.80). On average, participants took 8.09 minutes to complete the study (*S.D.* = 5.02).

**Procedure.** We followed a procedure similar to Study 1, except that we introduced six experimental conditions to which participants were randomly assigned: high anchor, low anchor, high-anchor consider-the-opposite, low-anchor consider-the-opposite, high-anchor mental-mapping, and low-anchor mental-mapping conditions. Study 2 used less-extreme values as anchors. A score of '2' on the cost dimension in the past supplier evaluation was used in the low-anchor conditions, while a score of '8' was used in the high-anchor conditions. The high-anchor and low-anchor conditions were identical to those used in Study 1. After reading the instructions for the evaluation tasks and the past evaluation scores of the supplier, in both the low-anchor consider-the-opposite and high-anchor consider-the-opposite conditions, participants were asked to give two reasons why the past evaluation score on the cost dimension was too low or high, respectively. Nagtegaal *et al.* (2020) [33] show that requiring two reasons is sufficient to reduce anchoring effects on managers' judgements in the public sector. For the low-anchor mental-mapping and high-anchor mental-mapping, the intervention was given *before* participants were presented with anchor values, consistent with the mental-mapping technique design by Bahník *et al.* (2019) [21]. The participants were asked to mentally map two reference points on a scale of 1–9. We asked them to *"please first estimate how well the best possible supplier performs in these criteria in your head, so the supplier should be given a score of 9 in all criteria. Next, do the same but for how badly the worst possible supplier performs in these four criteria in your head, so a score of 1 should be given in all criteria"*.

**Dependent measures.** The skewness and kurtosis coefficients for the dependent measures were computed across the six experimental conditions. All coefficients fell within the acceptable ranges, consistent with the criteria outlined in Study 1.

*Evaluation scores and the likelihood of using the same supplier in the future.* The same measures were adopted as in Study 1.

## Study 2 results

**Anchoring effects on multiple dimensional evaluations.** Univariate analyses of variance (ANOVAs) were used to test the effects of anchoring and debiasing techniques on the

evaluation scores in the four dimensions. Tables 1–4 report the related statistics of univariate analyses. The anchor has significant main effects on the evaluation of cost ($F(1,394) = 83.42$, $p <0.0005$), quality (F(1,394) = 4.07, p < 0.05), and service (F(1,394) = 12.16, p <0.01). However, we find no significant main effect of the anchor on the delivery evaluation scores (F (1,394) = 2.45, p = 0.12). Debiasing and interaction effects (anchoring × debiasing) have no significant main effects in any of the four dimensions (see Tables 1–4).

**Table 1. Effects of anchoring and debiasing techniques on evaluation of the cost dimension.**

| Source | SS | df | MS | F | p |
|---|---|---|---|---|---|
| Anchor | 166.01 | 1 | 166.01 | 85.59 | <0.0005 |
| Debiasing | 10.28 | 2 | 5.14 | 2.65 | 0.072 |
| Interaction | 7.08 | 2 | 3.54 | 1.82 | 0.16 |
| Explained | 184.00 | 5 | 36.80 | 18.97 | <0.0005 |
| Residual | 775.83 | 400 | 1.94 | | |
| Total | 12957.00 | 406 | | | |

*Note*. The anchor variable is binary, taking on values of either 0 or 1, with 0 indicating the low-anchor condition. The debiasing variable is categorical, taking on values of 0, 1 or 2. 0 indicates no debiasing technique; 1 represents the use of consider-the-opposite technique and 2 indicates the use of mental-mapping technique. This categorical distinctions remain consistent across Tables 2 to 5.

**Table 2. Effects of anchoring and debiasing techniques on evaluation of the quality dimension.**

| Source | SS | df | MS | F | p |
|---|---|---|---|---|---|
| Anchor | 10.27 | 1 | 10.27 | 4.57 | 0.033 |
| Debiasing | 11.56 | 2 | 5.78 | 2.57 | 0.078 |
| Interaction | 12.16 | 2 | 6.08 | 2.70 | 0.068 |
| Explained | 34.12 | 5 | 6.82 | 3.04 | 0.011 |
| Residual | 899.51 | 400 | 2.25 | | |
| Total | 15109 | 406 | | | |

**Table 3. Effects of anchoring and debiasing techniques on evaluation of the service dimension.**

| Source | SS | df | MS | F | p |
|---|---|---|---|---|---|
| Anchor | 27.35 | 1 | 27.35 | 11.19 | 0.001 |
| Debiasing | 13.51 | 2 | 6.76 | 2.76 | 0.064 |
| Interaction | 3.56 | 2 | 1.78 | 0.73 | 0.48 |
| Explained | 44.70 | 5 | 8.94 | 3.66 | 0.003 |
| Residual | 977.88 | 400 | 2.45 | | |
| Total | 11414 | 406 | | | |

**Table 4. Effects of anchoring and debiasing techniques on evaluation of the delivery dimension.**

| Source | SS | df | MS | F | p |
|---|---|---|---|---|---|
| Anchor | 7.26 | 1 | 7.26 | 3.22 | 0.074 |
| Debiasing | 12.94 | 2 | 6.47 | 2.87 | 0.058 |
| Interaction | 4.82 | 2 | 2.41 | 1.07 | 0.35 |
| Explained | 25.16 | 5 | 5.03 | 2.23 | 0.050 |
| Residual | 902.06 | 400 | 2.26 | | |
| Total | 16309 | 406 | | | |

**Table 5. Effects of anchoring and debiasing techniques on evaluation of the likelihood of choosing the same supplier.**

| Source | SS | df | MS | F | p |
|---|---|---|---|---|---|
| Anchor | 4404.85 | 1 | 4404.85 | 12.28 | 0.001 |
| Debiasing | 5228.55 | 2 | 2614.27 | 7.29 | 0.001 |
| Interaction | 2448.48 | 2 | 1224.24 | 3.41 | 0.034 |
| Explained | 12141.43 | 5 | 2428.29 | 6.77 | <0.0005 |
| Residual | 143445.09 | 400 | 358.61 | | |
| Total | 2113889 | 406 | | | |

Pairwise comparisons were conducted to examine whether the same anchoring effects found in Study 1 were replicated. Participants in the high-anchor condition rate the current performance in terms of cost more highly than those in the low-anchor condition ($M_{high\ anchor}$ = 6.21 vs. $M_{low\ anchor}$ = 4.58; $t(134)$ = 6.12, $p < 0.0005$, Cohen's d = 1.07). Next, we focused on the knock-on effects on the other dimensions. Participants in the high-anchor condition ($M_{high\ anchor}$ = 6.21) assign a higher quality evaluation than those in the low-anchor condition ($M_{low\ anchor}$ = 5.42; $t(134)$ = 2.80, $p < 0.01$, Cohen's d = 0.49). For the dimension of service, participants exposed to a high score on cost in the past evaluation rate service more highly than those exposed to a low score on cost ($M_{high\ anchor}$ = 5.28 vs. $M_{low\ anchor}$ = 4.56; $t(134)$ = 2.53, $p < 0.01$, Cohen's d = 0.44). Participants in the high-anchor condition ($M_{high\ anchor}$ = 6.22) also assign a higher evaluation to delivery than those in the low-anchor condition ($M_{low\ anchor}$ = 5.74; $t(134)$ = 1.74, $p < 0.05$, Cohen's d = 0.30). Overall, these results replicate the same pattern of anchoring effects when less extreme anchor values were used.

**Anchoring effect on the likelihood to use the same supplier in the future.** We also performed univariate analysis of the likelihood to continue the relationship with the supplier in the future. As shown in Table 5, significant main effects of anchoring ($F(1,394)$ = 11.16, $p < 0.01$) and debiasing ($F(1,394)$ = 6.80, $p < 0.01$) are observed. A significant interaction (anchoring × debiasing) is found for likelihood to use the same supplier in the future ($F(2,393)$ = 3.59, $p < 0.05$). This interaction effect suggests that anchoring differs depending on whether debiasing is at play and which debiasing strategy is employed (none, consider-the-opposite, or mental-mapping). The planned pairwise contrast reveals that participants in the high-anchor condition have a higher likelihood of using the same supplier in the future than participants in the low-anchor condition ($M_{high\ anchor}$ = 73.51% vs. $M_{low\ anchor}$ = 60.07%; $t(134)$ = 3.54, $p < 0.001$, Cohen's d = 0.61).

## Effectiveness of consider-the-opposite techniques

Tables 1–4 report the statistics of the univariate analyses regarding the effects of different debiasing techniques. First, we focused on the consider-the-opposite technique. A series of pairwise comparisons was conducted. Participants in the high-anchor consider-the-opposite condition report significantly lower evaluations of cost ($M_{high\ anchor\ opposite}$ = 5.75 vs. $M_{high\ anchor}$ = 6.21; $t(135)$ = -1.95, $p < 0.05$, Cohen's d = -0.33) and quality ($M_{high\ anchor\ opposite}$ = 5.75 vs. $M_{high\ anchor}$ = 6.21; $t(135)$ = -1.79, $p < 0.05$, Cohen's d = -0.31) than those in the high-anchor condition. However, we find no differences in the service score ($M_{high\ anchor\ opposite}$ = 5.09 vs. $M_{high\ anchor}$ = 5.30; $t(132)$ = -0.74, $p = 0.23$) and quality score ($M_{high\ anchor\ opposite}$ = 6.10 vs. $M_{high\ anchor}$ = 6.22; $t(132)$ = -0.44, $p = 0.33$). Thus, Hypothesis 4b is partially supported. Participants in the high-anchor consider-the-opposite condition have a significantly lower likelihood to use the same supplier in the future compared to those in the high-anchor condition ($M_{high\ anchor\ opposite}$ = 68.18% vs. $M_{high\ anchor}$ = 73.51%; $t(134)$ = -1.77, $p < 0.05$, Cohen's d = -0.31). These results support Hypothesis 4d.

Hypothesis 4a contends that the consider-the-opposite technique increases the evaluation scores of the low-anchor participants. However, pairwise comparisons show no significant differences in cost ($M_{low\ anchor\ opposite}$ = 4.58 vs. $M_{low\ anchor}$ = 4.76; $t(134)$ = 0.73, $p$ = 0.23), quality ($M_{low\ anchor\ opposite}$ = 5.78 vs. $M_{low\ anchor}$ = 5.41; $t(134)$ = 1.34, $p$ = 0.09), service ($M_{low\ anchor\ opposite}$ = 4.81 vs. $M_{low\ anchor}$ = 4.54; $t(134)$ = 1.02, $p$ = 0.16), and delivery scores ($M_{low\ anchor\ opposite}$ = 6.10 vs. $M_{low\ anchor}$ = 5.74; $t(134)$ = 1.41, $p$ = 0.08) between the low-anchor and the low-anchor consider-the-opposite conditions. These findings do not support Hypothesis 4a. Participants in the low-anchor consider-the-opposite condition have no significantly higher likelihood to use the same supplier in the future compared to those in the low-anchor condition ($M_{low\ anchor\ opposite}$ = 65.93% vs. $M_{low\ anchor}$ = 60.07%; $t(134)$ = 1.56, $p$ = 0.06). This finding does not support Hypothesis 4c.

## Effectiveness of mental-mapping techniques

Tables 1–5 report the statistics of the univariate analyses regarding the effects of mental-mapping. As predicted in Hypothesis 5a, participants in the low-anchor mental-mapping conditions evaluate cost performance ($M_{low\ anchor\ mapping}$ = 5.04 vs. $M_{low\ anchor}$ = 4.57; $t(134)$ = 2.04, $p < 0.05$, Cohen's d = 0.34), quality performance ($M_{low\ anchor\ mapping}$ = 6.04 vs. $M_{low\ anchor}$ = 5.41; $t(134)$ = 2.54, $p < 0.01$, Cohen's d = 0.44), service performance ($M_{low\ anchor\ mapping}$ = 5.09 vs. $M_{low\ anchor}$ = 4.54; $t(134)$ = 1.96, $p < 0.05$, Cohen's d = 0.34), and delivery performance more highly ($M_{low\ anchor\ mapping}$ = 6.24 vs. $M_{low\ anchor}$ = 5.74; $t(134)$ = 1.93, $p < 0.05$, Cohen's d = 0.33) than participants in the low-anchor condition. These findings support Hypothesis 5a. In the low-anchor mental-mapping condition, participants indicate a significantly higher likelihood to contract the same supplier in the future than those under the low-anchor condition ($M_{low\ anchor\ mapping}$ = 72.49% vs. $M_{low\ anchor}$ = 60.07%; $t(134)$ = 3.43, $p < 0.001$, Cohen's d = 0.59). Thus, Hypothesis 5c is supported.

Finally, we analysed the effectiveness of mental-mapping on high anchors. We find no significant differences between the high-anchor mental-mapping and the high-anchor conditions in the evaluations of cost ($M_{high\ anchor\ mapping}$ = 6.25 vs. $M_{high\ anchor}$ = 6.21; $t(134)$ = 0.18, $p$ = 0.43), quality ($M_{high\ anchor\ mapping}$ = 6.24 vs. $M_{high\ anchor}$ = 6.21; $t(134)$ = 0.11, $p$ = 0.46), service ($M_{high\ anchor\ mapping}$ = 5.59 vs. $M_{high\ anchor}$ = 5.28; $t(134)$ = 1.08, $p$ = 0.14), and delivery ($M_{high\ anchor\ mapping}$ = 6.57 vs. $M_{high\ anchor}$ = 6.22; $t(134)$ = 1.35, $p$ = 0.09). These findings do not support Hypothesis 5b. In the high-anchor mental-mapping condition, the likelihood of using the same supplier in the future is not statistically different from that under the low-anchor condition ($M_{high\ anchor\ mapping}$ = 76.59% vs. $M_{high\ anchor}$ = 73.51%; $t(134)$ = 0.99, $p$ = 0.16). These findings do not support Hypothesis 5d.

## Discussion

A growing body of behavioural operations research has begun to examine cognitive bias in decision-making [2,5,10,15,20,31,74,75]. However, the role of anchoring in multi-dimensional evaluation and supplier selection is underexplored. The present research focuses on how a supplier's past performance in one performance dimension anchors managers' assessment of a supplier's current performance across multiple dimensions. We also investigate how anchors impact the likelihood of using the same supplier in the future. To our knowledge, no previous research has studied the knock-on effects of anchoring and how these effects may be mitigated in multidimensional supplier evaluation. This study's theoretical and practical implications are particularly relevant because supplier performance largely determines the quality of products, which subsequently impacts the profitability of buying firms.

In Studies 1 and 2, the anchors are not indicative of current supplier performance. However, the results show that a supplier's past performance in one dimension (i.e., the cost) anchors participants' subsequent judgements about the current supplier performance in the corresponding dimension and other independent dimensions (i.e., quality, service, and delivery). Specifically, in both studies, participants exposed to a high (low) score on the supplier's past performance rate the same supplier more (less) favourably in all four dimensions. A high anchor also increases the likelihood that the same supplier is used in the future compared to a low anchor. The same pattern of results is observed when we employ less extreme values of high and low anchors, strengthening the anchoring effects. Together, our findings make an original contribution to production research, by showing how past supplier performance impacts managers' decisions in different aspects.

This research considers two different low-cost debiasing techniques. These techniques show asymmetrical effectiveness in reducing the high-anchor and low-anchor bias. The consider-the-opposite approach reduces the anchoring effects by lowering *some* of the evaluations and the likelihood of using the same supplier in the future when participants are presented with a high anchor. This technique reduces the anchoring bias and its knock-on effects on some of the performance dimensions. However, it does not decrease the power of a low anchor. Conversely, the mental-mapping technique helps mitigate the effects of a low anchor. When faced with two extreme reference points on an evaluation scale, those presented with a low anchor assign higher evaluation scores to the supplier in all dimensions and exhibit a higher likelihood of choosing the same supplier in the future. This result implies that high and low anchors require different debiasing techniques.

Several factors may explain the asymmetrical effectiveness of the two debiasing techniques. First, these findings may depend on the range of market prices provided in the experimental materials. For example, the cost of £3.0 is close to the lower end of the market price, ranging between £2.9 and £3.2. Second, participants may perceive this value to be consistent with the high evaluation scores received by the supplier in the past. Therefore, an increased evaluation score in the current cost evaluation may be justified in both the high-anchor and high-anchor mental-mapping conditions. Previous research on positive confirmation bias supports this explanation (see [76,77], for example). Conversely, in the consider-the-opposite scenario, participants are *instructed* to provide two reasons, potentially reducing the positive confirmation bias. The second reason is that more than one mechanism may contribute to the anchoring effects. In addition to the scale distortion theory of anchoring [23,24], other mechanisms may account for anchoring effects, such as the selective accessibility model [35,36,64,78] and insufficient adjustment of judgment due to an anchor [22]. This result confirms that different techniques are required to reduce the high-anchor and low-anchor effects.

This research provides a crucial theoretical contribution that the effectiveness of the debiasing techniques demonstrated in this research may be applied to various anchoring effects in other business and supply chain decision-making contexts, such as the newsvendor problem [25,28], the supplier evaluation that considers environmental sustainability [79,80] and bargaining in a multi-tier supply chain [52]. Thus, our findings pave the way for future research on debiasing anchoring effects. Furthermore, the efficacy of the mental-mapping technique, demonstrated in mitigating the influence of the low anchor, offers additional validation for scale distortion theory. This discovery bolsters the explanatory power of scale distortion theory in elucidating anchoring effects observed in psychological studies. Another significant contribution to research on judgment and decision-making is the development of de-biasing techniques within this field. These techniques hold potential for application in future anchoring studies, whether utilising direct or indirect comparative approaches. Such integration is likely to enhance the generalisability of our findings.

The findings of two experiments yield significant practical implications, suggesting that supply managers must exercise caution regarding the information received concerning a supplier's past performance. The presence of anchoring and knock-on effects implies that a supplier initially underperforming but subsequently improving may still be subject to biased evaluation. Consequently, it may prove challenging for such a supplier to impress the buying firm's manager if previous performance in one criterion fell short. Conversely, supply managers are predisposed to maintain relationships with suppliers who have demonstrated past success, even if their current performance is lacking. This assertion aligns with remarks from senior management during the company interview, noting that the sourcing and quality assurance team appeared to overly rely on suppliers' historical performance. Given the asymmetrical efficacy of debiasing techniques, managerial training should specifically address how to address anchoring effects when dealing with suppliers who have previously excelled or disappointed.

A limitation of our research pertains to the potential bounded effectiveness of the consider-the-opposite technique in real-life settings. For instance, when the same manager conducts a previous evaluation, they may be hesitant to disclose why the evaluation scores from the last round could be erroneous. Future research ought to examine this potential effect. Another limitation lies in the controlled amount of information provided to participants in this study. In actual managerial scenarios, managers may encounter information from diverse sources, which could serve as additional anchors. Future studies should investigate whether these potential anchors influence managers' decision-making processes.

On another note, within the experimental setup, cost performance indicators were presented in a range, while other performance indicators were expressed in absolute values, reflecting the practice of the company under study. The utilisation of a range as the performance indicator could influence judgement and evaluation. Although this practice remained consistent across all experimental conditions, and participants were randomly assigned to these conditions, it warrants consideration whether this might impact the findings in future research.

Finally, in the evaluation task, the cost attribute was positioned in the first row. While this arrangement might potentially influence results, it remained consistent across all experimental conditions. As only the absolute differences between conditions were compared, this setup aimed to minimise the impact of the cost attribute in the first row. Future studies in the realms of BOM, JDM, and Psychology should delve deeper into this potential effect across various contexts (e.g., quantity estimation).

In conclusion, the past performance of a supplier in one dimension has implications for current evaluations across multiple dimensions. As elucidated, managerial training should be tailored to explicitly tackle anchoring effects when engaging with suppliers who have previously demonstrated either excellence or disappointment. Furthermore, the findings underscore the necessity of employing distinct de-biasing techniques when addressing high and low anchor scenarios.

## Supporting information

**S1 File. S1 Study 1 data file.** This is the dataset of Study 1.
(SAV)

**S2 File. S2 Study 2 data file.** This is the dataset of Study 2.
(SAV)

## Acknowledgments

I hereby confirm that the data collection was approved by the University of committee of research ethics and that I comply with the journal's ethics and integrity policies. This research is partially supported by the Research Matching Grants Council of the Hong Kong Special Administrative Region (project: 700006 Applications of SAS Viya in Big Data Analytics), by providing the space and computer where I could conduct some data analyses during my summer break in Hong Kong.

## Author Contributions

**Data curation:** Ricky S. Wong.

**Formal analysis:** Ricky S. Wong.

**Methodology:** Ricky S. Wong.

**Software:** Ricky S. Wong.

**Validation:** Ricky S. Wong.

**Writing – original draft:** Ricky S. Wong.

**Writing – review & editing:** Ricky S. Wong.

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
