## [Decision Letter · Decision Letter 0]

28 Feb 2024

PONE-D-23-32273

The power of past performance in multidimensional supplier evaluation and supplier selection: Debiasing anchoring bias and its knock-on effects

PLOS ONE

Dear Dr. Wong,

Thank you for submitting your manuscript to PLOS ONE. After careful consideration, we feel that it has merit but does not fully meet PLOS ONE’s publication criteria as it currently stands. Therefore, we invite you to submit a revised version of the manuscript that addresses the points raised during the review process.

We look forward to receiving your revised manuscript.

Kind regards,

Ricardo Limongi

Academic Editor

PLOS ONE

Journal Requirements:

2. Please note that your Data Availability Statement is currently missing the repository name and/or the DOI/accession number of each dataset OR a direct link to access each database. If your manuscript is accepted for publication, you will be asked to provide these details on a very short timeline. We therefore suggest that you provide this information now, though we will not hold up the peer review process if you are unable.

Additional Editor Comments:

For this new version of the manuscript, we ask that you please send us a response letter to the reviewers and the editor indicating a response given to each of the items raised in the review.

If there is any comment about what you do not agree with, please provide us with a rationale for this.

Reviewers' comments:

Reviewer's Responses to Questions

**Comments to the Author**

1. Is the manuscript technically sound, and do the data support the conclusions?

Reviewer #1: Yes

Reviewer #2: Partly

2. Has the statistical analysis been performed appropriately and rigorously? 

Reviewer #1: Yes

Reviewer #2: Yes

3. Have the authors made all data underlying the findings in their manuscript fully available?

Reviewer #1: No

Reviewer #2: Yes

4. Is the manuscript presented in an intelligible fashion and written in standard English?

Reviewer #1: Yes

Reviewer #2: Yes

5. Review Comments to the Author

Reviewer #1: This research examines how anchoring bias affects managers' assessments of supplier performance and supplier selection and examines the effectiveness of two debiasing techniques.

Regarding the methodology used, the author carries out a consistency analysis of the proposed questionnaires. In study 1, independent sample t-test is applied to verify the relationship between anchoring and supplier evaluation. Study 2 was a between-subjects online factorial experiment, and applies ANOVA to analyze the effects of two debiasing techniques: consider-the-opposite and mental remapping.

The work is well structured and well-founded, however, I make some suggestions for improving the article.

1. The summary is good, but its structure could be improved, as the author presents the objective of the research, addresses the methodology, presents some results and talks again about the research topic. I suggest that the author follows a structured order to present the study summary, presenting the research topic, the question that the work intends to answer, the methodology used and finally the results obtained. Furthermore, the summary only mentions Study 1 and Study 2, but does not report the methodology used in these studies. I suggest that the methodology is better reported in the summary, avoiding using acronyms (N).

2. Regarding the results, although the author cites five tables, they are not included in the manuscript, making their analysis impossible.

3. The methods were detailed, however it was not explained what types of information about the suppliers' current performance the participants had to carry out the evaluation. This information appears only in the appendices. Given the importance of this data for the replication of the study, I suggest that it be better detailed in the text.

4. I suggest further discussion on whether a high anchor supplier's high current performance score is related to bias rather than current performance. What about the low current performance score of a supplier with a low anchor being related to bias and not current performance. Make it clearer to the reader how the questions in the questionnaires guarantee this relationship.

5. The statistical analysis is adequate. However, I suggest that the author better describes the conduct of statistical analyses, such as normality and t-test assumptions, to provide the reader with a better understanding of the characteristics of the samples. Furthermore, I suggest that the statistical software, version and packages used for statistical analyzes are described.

6. The data presented by the author support the conclusions. The questionnaires used are presented, which allow the study to be replicated in other contexts. However, I suggest that access to the questionnaire responses be made available so that the present study can be replicated.

7. All references are cited in the text and all citations are listed under references. However, it is necessary to check whether the citations are in accordance with the the submission guidelines, specifically in lines 233, 234, 250, 422 and 426.

Reviewer #2: When I saw the paper's title, I was looking forward to reading “The power of past performance in multidimensional supplier evaluation and supplier selection: Debiasing anchoring bias and its knock-on effects” by Ricky Siu Wong, PhD.

The experiment is basically successful and offers interesting results. However, I (currently still) have major reservations about whether these results offer any relevant added value for managers in the context of supplier evaluation.

The topic of the article is highly interdisciplinary. Relevant areas include psychology, judgment and decision-making, and supplier evaluation. In my opinion, all sub-areas should be adequately considered. However, my impression is that in this article, this is only the case to a very limited extent in the third area - supplier evaluation:

1.) The article does not take up the latest work in the field of behavioral OR,

2.) the motivation (in your Realism Check) is based on anecdotal relevance and was not derived from the literature and

3.) the evaluation problem, central to the studies, appears to be relatively remote from practice.

I believe the author is able to solve these issues. Therefore, I recommend a major revision of the paper.

In the following, you will find some ideas/questions/thoughts. Please refer to your text if I comment on something you have addressed somewhere in the script (and I did not find it).

1.) Could you please provide a graphical illustration of the studies, including all relevant information, to understand them?

2.) What was the time interval between the individual activities? Was there a distractor task? If not, why not?

3.) Please explain why you chose 104 or 408 participants.

4.) You write in the description for Study 1, “Prior to the task, there was an attention filter question ensuring that participants paid attention to the information provided. They were asked to indicate ‘strongly agree’ in the attention filter question.” Please provide information on how many people were screened out. What about Study 2? I cannot find information about an attention check.

5.) You write concerning the completion time “The mean completion time was 7.24 minutes (S.D. = 9.16). How much time die the faster 50% spend on average”. The SD seems high. An explanation might be that several participants have interrupted their assignment. Could you please provide in your answer to the reviewers a distribution of the completion time? (and if necessary, please comment on speeders and how you have dealt with them)

6.) I think it is fine. However, please explain why you chose a 9-point scale (and why you chose only to name the extremes)

7.) In the design of your supplier evaluation, many things must be considered (and commented). You do not have to address all of them in this paper. Some of my comments might also be useful for further research:

a. You write, “Study 1 presented a realistic multidimensional supplier evaluation task.” I have some doubts. Why should cost be evaluated on a Likert Scale in practice? Costs can be measured in USD, EUR, etc.

b. Why did you put your anchors on the dimension “costs”? Costs are often seen as the most crucial dimension.

c. Why did you put your anchors on the dimension in the first row? This might bias the participants.

d. Your supplier is allocated in Toyko. This information might bias your participants. Why did you consider this information?

e. You provide in a note the bandwidth of the costs. Why did you do that? Why did you not consider this information for the other dimensions? (In decision-making making, the bandwidth effect has crucial implications)

f. How did you choose the values for the other dimensions?

I hope the authors take up the challenge. Good luck.

6. PLOS authors have the option to publish the peer review history of their article (what does this mean?). If published, this will include your full peer review and any attached files.

Reviewer #1: No

Reviewer #2: No

---

## [Author Response · Author response to Decision Letter 0]

30 Mar 2024

Reviewer #1: This research examines how anchoring bias affects managers' assessments of supplier performance and supplier selection and examines the effectiveness of two debiasing techniques.

Regarding the methodology used, the author carries out a consistency analysis of the proposed questionnaires. In study 1, independent sample t-test is applied to verify the relationship between anchoring and supplier evaluation. Study 2 was a between-subjects online factorial experiment, and applies ANOVA to analyze the effects of two debiasing techniques: consider-the-opposite and mental remapping.

The work is well structured and well-founded, however, I make some suggestions for improving the article.

My response: Thank you sincerely for dedicating your time and efforts to reviewing our manuscript. We greatly appreciate your insightful comments, all of which have been carefully addressed in the revised version of the manuscript.

1. The summary is good, but its structure could be improved, as the author presents the objective of the research, addresses the methodology, presents some results and talks again about the research topic. I suggest that the author follows a structured order to present the study summary, presenting the research topic, the question that the work intends to answer, the methodology used and finally the results obtained. Furthermore, the summary only mentions Study 1 and Study 2, but does not report the methodology used in these studies. I suggest that the methodology is better reported in the summary, avoiding using acronyms (N).

My response: Thank you very much for your time and invaluable feedback, which I truly appreciate. I have taken your suggestions into consideration and revised the summary/abstract accordingly. Additionally, I have included the methodology used in the abstract as per your recommendation.

2. Regarding the results, although the author cites five tables, they are not included in the manuscript, making their analysis impossible.

My response: Thank you for your message. I understand that the tables were included in the main text in this revision, specifically in the Result section of Study 2. I appreciate your effort to ensure accessibility by also attaching the tables in Appendix 1 at the end of the letter. Thank you again.

3. The methods were detailed, however it was not explained what types of information about the suppliers' current performance the participants had to carry out the evaluation. This information appears only in the appendices. Given the importance of this data for the replication of the study, I suggest that it be better detailed in the text.

My response: Thank you for your valuable comment. I wholeheartedly agree that replication of the study is crucial. In the revised version, I have addressed this concern by providing detailed illustrations of the evaluation process (please see lines 391-416) and the information pertaining to the suppliers' current performance in Figures 1-2 (please refer to lines 405-407 and fig1.tiff and fig2.tiff files). Furthermore, I have revised the methodology section to clarify the types of information about the supplier’s current performance provided to the participants for conducting the evaluation.

Moreover, I would like to highlight that the effects of past performance in one dimension on the current evaluations of suppliers’ performance in multiple dimensions have been successfully replicated in Study 2. As stated in the Results section of Study 2, “Participants in the high-anchor condition rate the current performance in terms of cost more highly than those in the low-anchor condition (Mhigh anchor = 6.21 vs. Mlow anchor = 4.58; t(134) = 6.12, p < 0.0005, Cohen’s d = 1.07).” . 

4. I suggest further discussion on whether a high anchor supplier's high current performance score is related to bias rather than current performance. What about the low current performance score of a supplier with a low anchor being related to bias and not current performance. Make it clearer to the reader how the questions in the questionnaires guarantee this relationship.

My response: Thank you once again for your valuable comment. I have duly noted your suggestion, and corresponding changes have been implemented on lines 347-355 and 424-422. If you have any further feedback or concerns, please do not hesitate to let me know. Your input is greatly appreciated.

5. The statistical analysis is adequate. However, I suggest that the author better describes the conduct of statistical analyses, such as normality and t-test assumptions, to provide the reader with a better understanding of the characteristics of the samples. Furthermore, I suggest that the statistical software, version and packages used for statistical analyzes are described.

My response: Thanks a lot. This is a great idea. I used SPSS version 29 (which has now been included in the manuscript). I completely followed your advice. And, the revised manuscript discussed and tested if the normality assumption was violated, by showing that both skewness coefficient is between ‐2 to +2 and kurtosis coefficient is between ‐7 to +7. These are consistent with the normality assumption suggested by Bryne (2010) [72] and Hair et al. (2010) [73]. The approach I have taken to test the normality assumption by examining the skewness and kurtosis coefficients is sound and aligns with the guidelines suggested by Bryne (2010) [72] and Hair et al. (2010) [73]. It is important to ensure that your data meet the normality assumption of independent samples t-test.

I also re-examine the data using a conservative approach of the Mann Whitney tests. the findings are consistent with the independent-sample t-tests and planned contrasts used in Studies 1 & 2. The Mann Whitney test results are included in Appendix 2 at the bottom of this response letter for your reference (but the Mann Whitney test results were not reported in the manuscript, because the skewness and kurtosis coefficients have shown that the data do not violate the independent sample t-test assumption). 

6. The data presented by the author support the conclusions. The questionnaires used are presented, which allow the study to be replicated in other contexts. However, I suggest that access to the questionnaire responses be made available so that the present study can be replicated.

My response: Thank you very much. I regret to inform you that the datasets were not locatable. It is imperative that findings from experimental studies are replicable, a sentiment with which I wholeheartedly agree. The datasets have now been made accessible via the PLOS One submission portal, enabling other scholars to access the questionnaire responses.

7. All references are cited in the text and all citations are listed under references. However, it is necessary to check whether the citations are in accordance with the submission guidelines, specifically in lines 233, 234, 250, 422 and 426.

My response: Thank you for bringing this to my attention, and I apologise for my oversight. In this revised version, all citations throughout the manuscript adhere to the PLOS One submission guidelines. Please refer to lines 47 and 51 as examples. 

Reviewer #2: When I saw the paper's title, I was looking forward to reading “The power of past performance in multidimensional supplier evaluation and supplier selection: Debiasing anchoring bias and its knock-on effects” by Ricky Siu Wong, PhD.

The experiment is basically successful and offers interesting results. However, I (currently still) have major reservations about whether these results offer any relevant added value for managers in the context of supplier evaluation.

My response: Thank you for dedicating your valuable time to provide such insightful feedback. This research is driven by both a theoretical gap, namely the anchoring effect in multi-dimensional evaluation and de-biasing methods, and a tangible issue faced by a prominent organisation. Upon publication, the organisation with whom we collaborated has agreed to implement the de-biasing methods developed in this research and assess their real-world impacts on their supplier evaluation processes. This collaboration holds the potential to yield a significant and impactful case study in the future.

It is noteworthy that both de-biasing techniques proposed in this research offer cost-effective decision aids for mitigating the effects of low and high anchors on supplier evaluation and selection processes. Additionally, an essential practical implication lies in the ripple effects of anchoring on other attributes under evaluation. In the revised manuscript, particular emphasis has been placed on highlighting these practical implications in both the Introduction and Discussion sections (see lines 70-75, 86-95, 107-112, 651-654, 663-666, 677-678, 695-700, 710-723).

Another significant practical contribution concerns the adaptation and modification of Bahník et al.'s (2019) debiasing method for use in multi-dimensional supplier evaluation. This study introduces a novel approach utilising the extreme points of the Likert scale, a method not previously tested in the literature. This innovative approach offers substantial added value for managers engaged in supplier evaluation or similar evaluative tasks across various domains. The Discussion section places particular emphasis on these practical implications (see lines 710-723).

R2’s comment: The topic of the article is highly interdisciplinary. Relevant areas include psychology, judgment and decision-making, and supplier evaluation. In my opinion, all sub-areas should be adequately considered. However, my impression is that in this article, this is only the case to a very limited extent in the third area - supplier evaluation:

My response: Thanks a lot for your feedback. Please note that psychology and judgment and decision-making (JDM) were indeed discussed in the initial paragraphs under "Theoretical Background and Hypotheses." Recent JDM literature, such as the work by Bahník et al. (2019) [21], was utilised to elucidate the underlying mechanisms through which anchoring influences judgments within a supplier evaluation context (please refer to lines 183-212). Furthermore, both this section and the Discussion have been revised to address your concern (see lines 687-694 for specific examples).

I greatly appreciate your invaluable feedback, which undoubtedly contributes to the strengthening of the paper. The literature section has been enhanced by incorporating JDM and psychology studies (please refer to lines 73-75, 136-211). Additionally, in the revised Discussion section, the relevance of our findings to the fields of JDM and psychology is expounded upon. For instance, the asymmetrical difficulties encountered in de-biasing high versus low anchors offer avenues for future research across all relevant disciplines. Future studies should consider employing a combination of de-biasing methods. The empirical findings from this research provide further support for the scale of distortion theoretical framework regarding the observed anchoring effect. Importantly, the findings suggest that low and high anchors may operate somewhat differently. To the best of my knowledge, this study represents the first investigation into whether these methods could mitigate anchoring's knock-on effects in multi-dimensional evaluation scenarios. These findings also pave the way for future research within the realms of JDM and psychology.

1.) The article does not take up the latest work in the field of behavioral OR,

My response: Thanks a lot for this important comment. I take your advice seriously and have thoroughly revised the entire manuscript to demonstrate a mastery of the literature. In doing so, the following behavioural studies have been discussed and cited:

7. Bendoly E, Donohue K, Schultz KL. Behavior in operations management: Assessing recent findings and revisiting old assumptions. Journal of operations management. 2006;24(6):737-52.

10. Perera HN, Fahimnia B, Tokar T. Inventory and ordering decisions: a systematic review on research driven through behavioral experiments. International Journal of Operations & Production Management. 2020;40(7/8):997-1039.

11. Bendoly E, Croson R, Goncalves P, Schultz K. Bodies of knowledge for research in behavioral operations. Production and operations management. 2010;19(4):434-52.

12. Croson R, Schultz K, Siemsen E, Yeo M. Behavioral operations: the state of the field. Journal of Operations Management. 2013;31(1-2):1-5.

13. Kull TJ, Oke A, Dooley KJ. Supplier selection behavior under uncertainty: contextual and cognitive effects on risk perception and choice. Decision Sciences. 2014;45(3):467-505.

14. Schiffels S, Fliedner T, Kolisch R. Human behavior in project portfolio selection: Insights from an experimental study. Decision Sciences. 2018;49(6):1061-87.

15. Tokar T, Aloysius JA, Waller MA. Supply chain inventory replenishment: The debiasing effect of declarative knowledge. Decision Sciences. 2012;43(3):525-46.

16. Yamini S. Behavioural operations management: trends and insights. Emerging Frontiers in Operations and Supply Chain Management: Theory and Applications. 2021:233-49.

17. Zhan Y, Chung L, Lim MK, Ye F, Kumar A, Tan KH. The impact of sustainability on supplier selection: A behavioural study. International Journal of Production Economics. 2021;236:108-18.

19. Moritz BB, Hill AV, Donohue KL. Individual differences in the newsvendor problem: Behavior and cognitive reflection. Journal of Operations Management. 2013;31(1-2):72-85.

20. Wong RS. An experimental investigation of attribute-framing effects on ordering decisions in dual sourcing: Role of attention to suppliers’ information. International Journal of Operations and Production Management. 2023.

25. D'Urso D, Di Mauro C, Chiacchio F, Compagno L. A behavioural analysis of the newsvendor game: Anchoring and adjustment with and without demand information. Computers & Industrial Engineering. 2017;111:552-62.

29. Davis AM, Hyndman K. Multidimensional bargaining and inventory risk in supply chains: An experimental study. Management Science. 2019;65(3):1286-304.

31. Doyle J, Ojiako U, Marshall A, Dawson I, Brito M. The anchoring heuristic and overconfidence bias among frontline employees in supply chain organizations. Production Planning & Control. 2021;32(7):549-66.

74. Schultz KL, Robinson LW, Thomas LJ, Schultz J, McClain JO. The use of framing in inventory decisions. Production and Operations Management. 2018;27(1):49-57.

75. Villa S, Castañeda JA. A behavioural investigation of power and gender heterogeneity in operations management under uncertainty. Management Research Review. 2020;43(6):753-71.

2.) the motivation (in your Realism Check) is based on anecdotal relevance and was not derived from the literature.

My response: I am sorry that I did not make it sufficiently clear in the original manuscript, and the motivation is also supported by the relevant literature and the real problems faced by a large organisation. The literature that supports my realism check has now been included in the relevant section (please see below for the citations).

20. Wong RS. An experimental investigation of attribute-framing effects on ordering decisions in dual sourcing: Role of attention to suppliers’ information. International Journal of Operations and Production Management. 2023.

65. Rungtusanatham M, Wallin C, Eckerd S. The vignette in a scenario‐based role‐playing experiment. Journal of Supply Chain Management. 2011;47(3):9-16.

66. Golgeci I, Ali I, Bozkurt S, Gligor DM, Arslan A. The impact of corporate support programs on environmental and social innovation: empirical insights from the food and beverage industry. International Journal of Operations & Production Management. 2022 (ahead-of-print).

67. Bachrach DG, Bendoly E. Rigor in behavioral experiments: A basic primer for supply chain management researchers. Journal of Supply Chain Management. 2011;47(3):5-8.

3.) the evaluation problem, central to the studies, appears to be relatively remote from practice.

My response: Thank you very much for your specific comment. As recommended by the literature [1,2,3,79], multi-dimensional evaluation is prevalent in supply chai

---

## [Decision Letter · Decision Letter 1]

30 Apr 2024

The power of past performance in multidimensional supplier evaluation and supplier selection: Debiasing anchoring bias and its knock-on effects

PONE-D-23-32273R1

Dear Dr. Wong,

We’re pleased to inform you that your manuscript has been judged scientifically suitable for publication and will be formally accepted for publication once it meets all outstanding technical requirements.

Kind regards,

Ricardo Limongi

Academic Editor

PLOS ONE

**Comments to the Author**

1. If the authors have adequately addressed your comments raised in a previous round of review and you feel that this manuscript is now acceptable for publication, you may indicate that here to bypass the “Comments to the Author” section, enter your conflict of interest statement in the “Confidential to Editor” section, and submit your "Accept" recommendation.

Reviewer #1: All comments have been addressed

Reviewer #2: All comments have been addressed

2. Is the manuscript technically sound, and do the data support the conclusions?

Reviewer #1: Yes

Reviewer #2: (No Response)

3. Has the statistical analysis been performed appropriately and rigorously? 

Reviewer #1: Yes

Reviewer #2: Yes

4. Have the authors made all data underlying the findings in their manuscript fully available?

Reviewer #1: Yes

Reviewer #2: Yes

5. Is the manuscript presented in an intelligible fashion and written in standard English?

Reviewer #1: Yes

Reviewer #2: No

6. Review Comments to the Author

Reviewer #1: (No Response)

Reviewer #2: Dear author,

I am pleased that you have made the effort necessary to write one that is worth reading. Please make sure that any linguistic errors are eliminated before the final printing.

7. PLOS authors have the option to publish the peer review history of their article (what does this mean?). If published, this will include your full peer review and any attached files.

Reviewer #1: No

Reviewer #2: No

---

## [Editor Report · Acceptance letter]

3 May 2024

PONE-D-23-32273R1 

PLOS ONE

Dear Dr. Wong, 

I'm pleased to inform you that your manuscript has been deemed suitable for publication in PLOS ONE. Congratulations! Your manuscript is now being handed over to our production team.

Kind regards, 

on behalf of

Professor Ricardo Limongi 

Academic Editor

PLOS ONE